# Parental Burnout, Negative Parenting Style, and Adolescents’ Development

**DOI:** 10.3390/bs14030161

**Published:** 2024-02-22

**Authors:** Xingchen Guo, Chenyu Hao, Wei Wang, Yongxin Li

**Affiliations:** Institute of Psychology and Behavior, Henan University, Kaifeng 475001, China; gxchen1992@163.com (X.G.); haochenyu0422@163.com (C.H.); 10030117@vip.henu.edu.cn (W.W.)

**Keywords:** adolescent development, negative parenting styles, parental burnout

## Abstract

Research on parental burnout has focused more on its antecedents than on its consequences. Burned-out parents may experience a series of behavioral changes, negatively affecting their children’s physical and mental development. This study examined the effects of primary caregivers’ parental burnout on adolescents’ development and the mediating role of negative parenting styles. This study used a time-lagged design, and data were collected at three different time points. Adolescents were asked to identify their primary caregivers, and parents were asked whether they were the primary caregivers of their children. Thereafter, paired data from the children and primary caregivers were collected. A total of 317 junior middle school students (178 boys, M_age_ = 14.20 ± 0.8 years) and primary caregivers (71 fathers, M_age_ = 42.20 ± 4.53 years) from Henan province participated. Primary caregivers’ parental burnout was positively associated with negative parenting styles, and negative parenting styles mediated the relationship between parental burnout and adolescent development. From the perspective of prevention-focused interventions, it is necessary to focus on preventing the occurrence of parental burnout. Further, parents should try to avoid using abusive behaviors toward their children and neglecting them.

## 1. Introduction

In modern times, parents may experience diminished efficiency and heightened parenting pressure compared to earlier generations [1]. When individuals are chronically exposed to the stresses of parenting and lack adequate parenting resources, they become susceptible to parental burnout [2]. Considering the limited attention given to the impacts of parental burnout on adolescents’ development, this study explored the effects of parental burnout in primary caregivers on adolescents’ development, as well as the mediating role of negative parenting styles.

Parenting plays a fundamental role in children’s development [3]. A positive parenting style can yield favorable developmental results, including the cultivation of healthy parent–child relationships, fostering a warm and loving family environment, and enhancing parental self-efficacy [4]. Conversely, embracing a negative parenting style could lead to detrimental consequences, as such parenting behaviors are associated with the development of strained parent–child relationships [5].

Parental burnout can serve as a significant antecedent to parenting style. This psychological syndrome arises from chronic parenting stress and a dearth of resources [2]. It encompasses a range of adverse symptoms, such as emotional exhaustion (e.g., feeling exhausted when waking up in the morning and contemplating the day’s interactions with the children), difficulty in experiencing a sense of achievement and happiness (e.g., sensing a decline in one’s parenting abilities), emotional detachment from one’s children (e.g., no longer putting in effort for the children beyond regular caregiving), and disenchantment with parental roles (e.g., finding it challenging to tolerate the demands of parenthood) [6]. According to the balance between risks and resources theory [2], parental burnout manifests when parents grapple with chronic parenting demands without sufficient resources for coping. As proposed by the theory, to mitigate further resource depletion, burned-out parents may be inclined to adopt neglecting or controlling behaviors toward their child [7,8], leading to the manifestation of a negative parenting style.

Previous studies have largely improved our understanding of the effects of parental burnout on children’s development; however, some questions remain to be further explored. First, studies that focused on the negative influence of parental burnout on children used only one or two indicators, which may limit our understanding of the complex consequences of parental burnout. For instance, parental burnout is positively associated with adolescents internalizing and externalizing problems [9,10] and academic burnout [11], while it is negatively related to adolescents’ academic performance and social adaptation [12]. To further explore the impact of parental burnout on adolescents’ development, this study chose the indicators of academic achievement and social adaptation from the work of Dong and Lin [13], and each indicator considered both positive and negative perspectives.

Second, how can parental burnout influence children’s development? Parental burnout is associated with partner estrangement and conflict, child neglect, and abuse [7,14]; thus, burned-out parents may adopt negative parenting styles or activities. For example, a negative parenting style may increase the internalization and externalization of problem behaviors in adolescents [15]. Therefore, a negative parenting style may mediate the relationship between parental burnout and children’s development. Particularly, psychological control and parental neglect have been extensively studied in prior studies [16,17]. Further, psychological control and parental neglect may exhibit correlations with parental burnout, potentially exerting direct predictive influences on the physical and mental development of adolescents [18]. Consequently, psychological control and parental neglect were chosen as mediators in this study.

Third, previous studies have focused solely on either the father or mother of a given family. While the responsibility of raising children is taken on by mothers in most families, in some families, fathers may also take on these responsibilities. This may suggest that the primary caregivers may be different in different families, the role of primary caregivers may be greater than that of secondary caregivers, and parental burnout may have a more important impact on children’s growth. Therefore, this study focused on the perception of primary caregivers (the person in the family, from the perspective of adolescent, who is taking the main responsibility of taking care of them) and examined the effect of parental burnout on children’s development.

As children become more autonomous and exert greater influence on the family and the parent–child relationship when they are in adolescence [19], children may begin to have active roles in families rather than being passive recipients of their parents’ caregiving behaviors [20]. With rapid physical maturity and the relatively slow development of emotional maturity [21], adolescents may experience various problems like increasing conflict with their parents or other behavioral and emotional concerns, which may lead to a higher level of parental burnout. Moreover, adolescents’ parents are often confronted with new parenting challenges, like the greater autonomy needs of adolescents, which may be related to higher levels of parenting stress [22]. Therefore, this study focuses on a sample of adolescents and their primary caregivers.

According to the balance between risks and resources (BR2) model [2] of parental burnout, burned-out parents may already be exhausted and lack parenting resources to cope with their children’s needs. They may adopt more negative parenting behaviors (e.g., neglect and control) to save their own resources, and these negative parenting behaviors will further affect children’s healthy development. Based on the identified primary caregivers, this study examined the effects of primary caregiver parents on their children’s health development and the mediating effects of negative parenting styles. Psychological control and parental neglect were specifically selected as indicators of negative parenting styles. Psychological adaptation, friendship quality, and academic engagement were selected as indicators of adolescent development; the indicators were taken from both positive and negative perspectives. The framework of this study is illustrated in Figure 1.

### 1.1. Adolescents’ Development Indicators

As noted above, a more integrated perspective on adolescent development is needed to further clarify the negative effects of parental burnout on adolescent development. Dong and Lin [13] provided the foundational work in a seminal study on the primary psychological developmental indicators of Chinese children and adolescents. Leading a team of researchers, they conducted surveys involving over 100 thousand adolescents across diverse regions of China. Their research not only established a comprehensive system of key psychological developmental indicators for Chinese children and adolescents but also curated a robust database and set of norms. According to Dong and Lin [13], the developmental indicators of Chinese adolescents can be categorized into four aspects: cognitive ability, academic achievement, social adaptation, and developmental environment. As cognitive ability was relatively stable, developmental environment was not as distal an outcome as other indicators. This study focused on academic achievement and social adaptation. Psychological adaptation and friendship quality were specifically selected as indicators of social adaptation, while academic engagement was selected as an indicator of academic achievement.

More specifically, psychological adaptation refers to the state of physical and mental stability, harmony, and balance of an individual that is formed during the process of interacting with the environment through self-regulation, which is an outward manifestation and an important symbol of mental health [23]. Friendship quality is defined as the quality characteristics of an individual’s friendship with peers and is an important index for measuring adolescents’ interpersonal communication and social adaptability [24]. Academic engagement is the external manifestation of students’ internal motivation to participate in school education. It is also an important predictor of adolescent academic performance [25].

As the indicators may not be a one-dimensional concept, each key indicator was examined from both a positive and a negative approach. Zou et al. [26] indicated that adaptive functioning could be evaluated from two aspects: adaptation and maladaptation. Paker and Asher [27] showed that friendship quality was not a one-dimensional concept and had positive and negative aspects. Skinner et al. [28], based on the internal motivation of individuals, defined academic engagement as behavioral and emotional engagement (positive) or disaffection (negative). In sum, positive and negative psychological adaptation, positive and negative friendship quality, academic engagement, and academic disaffection were selected as indicators of adolescent development in this study.

### 1.2. Parental Burnout and Adolescent Development

Parents may develop burnout when they are extremely exhausted and lack the resources to cope with parenting stress [2]. Consequently, adolescents cannot obtain adequate support and resources from their parents. The lack of support and resources may eventually affect adolescents’ interpersonal, social, and academic development. In addition, burned-out parents tend to distance themselves from their children emotionally and behaviorally and lack the energy to continuously spend resources on their children’s psychological needs [14]. This may inhibit the ability of adolescents to obtain emotional support from their parents. As adolescents confront rapid changes, without enough support and guidance from their parents, they may be prone to negative thoughts and experiences about themselves, others, and their surrounding environment, which may affect their social development. Parental burnout is negatively associated with adolescents’ mental health [29] and social adaptation [12], while positively associated with problem behaviors [9]. Thus, parental burnout may have a direct effect on adolescent development. In line with prior studies, Hypothesis 1 was proposed:

**H1:** *Parental burnout is negatively associated with adolescents’ positive development and positively associated with adolescents’ negative development*.

### 1.3. Parental Burnout and Negative Parenting Styles

Parenting styles refer to a series of strategies and methods used in the process of parenting, which are comprehensive expressions of parents’ values, attitudes, and behaviors regarding parenting activities [30], like positive parenting styles (e.g., warmth, support, and respect) and negative parenting styles (e.g., overprotection, rejection, and neglect). Long-term parenting stress changes parenting attitudes [31], and parental burnout is positively correlated with negative parenting styles [32]. Burned-out parents are more likely to maintain emotional distance from their children and exhibit more abuse and neglectful behaviors [7]. Therefore, parental burnout may increase parents’ negative parenting style.

Negative parenting styles comprise variable behaviors and styles like rejection, overprotection, and punishment [33]. Among these, psychological control and parental neglect have been frequently studied in prior studies. Specifically, psychological control is defined as parents’ manipulation of their children’s thoughts and feelings by inducing guilt, love withdrawal, and excessive personal control [16]. Parental neglect is regarded as behaviors in which parents neglect the basic needs of their children for a long time (including emotional, physical, supervisory neglect, and educational neglect) [17]. To avoid the continuous consumption of parenting resources, burned-out parents who find it difficult to gain happiness and achievement from parent–child interactions may deliberately ignore their children’s developmental needs [7] and tend to control their children’s emotions and behaviors by emphasizing authority, expressing disappointment, and inducing guilt [8]. Parental burnout is positively correlated with both psychological control and neglect [14,34]. Therefore, psychological control and parental neglect are regarded as measurement indicators of a negative parenting style, and thus, Hypothesis 2 was proposed:

**H2:** 
*Parental burnout is positively associated with negative parenting styles (psychological control and parental neglect).*


### 1.4. Mediating Effect of Negative Parenting Styles

An ideal parenting style is a demonstration of a stable attitude and behavior by parents toward the growth of their children in daily life [35]. “Teach by personal example as well as verbal instruction,” parenting styles are imperative to laying the foundation for the growth of adolescents. The extant literature has indicated that parenting styles are closely related to adolescents’ self-acceptance, mental health, and academic achievement [36,37]. Further, adolescents are easily influenced by their parents’ negative parenting styles, which hinder the formation of their self-identity and negatively affect their cognitive and behavioral development [15]. Thus, the negative effects of parental burnout may not be confined to the caregivers themselves; they may also spill over into their children. Specifically, when parents with high levels of parental burnout adopt excessively negative parenting styles, adolescents cannot obtain the developmental resources they need from their parents. Therefore, they are likely to develop negative emotions like anxiety and depression [29], ultimately affecting their physical and mental development.

In summary, the following hypothesis was proposed:

**H3:** 
*A negative parenting style mediates the relationship between parental burnout and adolescents’ positive and negative development.*


## 2. Materials and Methods

### 2.1. Participants and Procedures

The sample comprised junior school students and their primary caregivers (father or mother) from Henan province. In mitigating the impact of common method bias [38], data were collected on three different occasions with an interval of one month through convenience sampling. Parents were asked to indicate parental burnout at Time 1. Students were asked to indicate their perceived parental psychological control and parental neglect at Time 2, and psychological adaptation, friendship qualities, and academic engagement at Time 3. The researchers distributed the sealed questionnaire to students in class. The students were required to deliver it to their primary caregivers for the latter to complete. The students then had to return the sealed questionnaires. The second and third surveys were conducted during lectures. Participation was voluntary, and they could withdraw at any time. This survey was approved by the research ethics committee of the authors’ academic institution.

A total of 453 questionnaires were distributed, of which 436 were returned at Time 1. After eliminating invalid questionnaires, 426 valid questionnaires were obtained (effective return rate = 94.04%). A total of 447 and 443 student questionnaires were collected at Times 2 and 3, respectively. After matching with the parents’ questionnaires, 387 groups of effective paired data were obtained.

The “primary caregiver” item was constant in both the parents’ and students’ versions of the questionnaire. Parents were asked whether they were the primary caregiver, and students were asked which of their parents they considered to be the primary caregiver. Seventy pairs of data were removed because parents had different responses regarding who was the primary caregiver of their children. A total of 317 paired data points remained (parents, M_age_ = 42.20 ± 4.53 years; adolescents, M_age_ = 14.20 ± 0.8 years). Regarding primary caregivers, there were 246 mothers (77.60%), and 71 were fathers (22.40%). Regarding educational levels, 243 (76.70%) completed less than high school education, and 64 (20.10%) completed more than undergraduate education. This study comprised 178 (56.20%) boys and 139 (43.80%) girls.

To calculate the statistical power of the hypothesized model, power testing was conducted based on the model fit proposed by Preacher and Coffman [39]. The significance was set to 0.05, the degree of freedom was set to 29, the sample size was 317, the null root mean square error of approximation (RMSEA) was 0, and the alternative hypothesized RMSEA was set to 0.08. The power of the model was 0.99, suggesting the sample size was sufficient.

Because of the involvement of human participants, this study underwent a review and received approval from the Research Ethics Committee of the Institute of Psychology and Behavior, Henan University (no. 20190329001). Written informed consent was obtained from participants to publish this study. Participants were presented with the following instructions: “Please check this box if you agree that the data can be used for academic research and publication”.

### 2.2. Measures

#### 2.2.1. Parental Burnout

Parental burnout was measured using the Chinese version of the Parental Burnout Assessment [40]. It comprises seven items, and each item is rated using a seven-point Likert scale ranging from one (completely inconsistent) to seven (completely consistent), with a higher score representing higher burnout. An example is “I feel like I can’t cope as a parent”. In this study, Cronbach’s α was 0.88, indicating good internal consistency.

#### 2.2.2. Parental Psychological Control

Parental psychological control was measured using the Parental Psychological Control Questionnaire [41]. It comprises 18 items, and each item is rated on a five-point Likert scale ranging from one (completely inconsistent) to five (completely consistent), with a higher score representing higher perceived parental psychological control. An example is “When I did not live up to my parents’ expectations, they told me I should feel guilty”. In this study, Cronbach’s α was 0.93, indicating excellent internal consistency.

#### 2.2.3. Parental Neglect

Parental neglect was measured using the Neglect Scale of the Child Psychological Neglect Scale [42]. It comprises 17 items, and each item is rated on a five-point Likert scale ranging from one (never) to five (always), with a higher score representing higher perceived parental neglect. An example is “Parents didn’t help me when I don’t understand something”. In this study, Cronbach’s α was 0.89, indicating good internal consistency.

#### 2.2.4. Adolescent Psychological Adaptation

Adolescent psychological adaptation was measured using the Strengths and Difficulties Questionnaire [43] that comprises 25 items measured in two sections: strengths (e.g., “I try to be friendly to other people, and I care about people’s feelings”) and difficulties (e.g., “I feel very angry and often lose my temper”), and each item was rated using a three-point Likert scale ranging from 0 (completely inconsistent) to 2 (completely consistent). In this study, the strengths section was regarded as relating to positive psychological adaptation, whereas the difficulties section was regarded as relating to negative psychological adaptation, with a higher score representing higher positive/negative psychological adaptation. The Cronbach’s α values for positive and negative psychological adaptation were 0.70 and 0.78, respectively, indicating acceptable internal consistency.

#### 2.2.5. Adolescent Friendship Quality

Adolescent friendship quality was measured using a shortened Chinese version of the Friendship Quality Questionnaire [44], which comprises 18 items. Each item was rated on a five-point Likert scale ranging from 0 (completely inconsistent) to 4 (completely consistent). In this study, the Conflict and Betrayal section was regarded as relating to negative friendship quality (e.g., “We are often angry with each other”), while the other five sections were regarded as positive friendship qualities (e.g., “We always discuss our problems together”), with a higher score representing higher negative/positive friendship qualities. The Cronbach’s α values for positive and negative friendship qualities were 0.80 and 0.56, respectively.

#### 2.2.6. Adolescent Academic Engagement

Adolescent academic engagement was measured using the Behavior and Emotional Engagement and Disaffection Questionnaire [45], which comprises 20 items measured in four sections: Behavior Engagement, Emotional Engagement, Behavior Dissection, and Emotional Dissection. Each item is rated on a four-point Likert scale ranging from 1 (completely inconsistent) to 4 (completely consistent). In this study, the Behavior Engagement and Emotional Engagement sections were regarded as relating to academic engagement (e.g., “I concentrate on my lessons”), while the Behavior Disaffection section and Emotional Disaffection section were regarded as relating to academic disaffection (e.g., “I feel terrible in class”), with a higher score representing higher academic engagement/disaffection. The Cronbach’s α values for academic engagement and disaffection were 0.82 and 0.91, respectively, indicating good internal consistency.

### 2.3. Data Analysis

The original data were analyzed using SPSS 21.0 and AMOS 24.0 (IBM, Armonk, NY, USA). First, common method variance was examined using an unmeasured latent method factor. Second, descriptive statistics and correlation analyses were performed. Third, the hypothesized comprehensive hypothesis model was examined using structural equation modeling.

## 3. Results

### 3.1. Common Method Bias

Common method variance was examined by controlling for the effects of the unmeasured latent method factor [38]. Using the confirmatory factor analysis model, each item was loaded on its respective construct (i.e., parental burnout, parental psychological control, parental neglect, psychological adaptation, friendship quality, or academic engagement). Moreover, all items were loaded on another latent variable, and the path coefficients were set to be equal. The variance explained by the latent method factor was 0.76%, which is lower than the median of 25% reported in a previous study [46]. These results provide evidence that common method bias did not significantly bias the results.

### 3.2. Descriptions and Correlations

Table 1 shows the descriptive statistics and Pearson’s correlations for all measured variables. As shown, parental burnout only showed significant correlations with psychological control (r = 0.19, *p* < 0.01), academic engagement (r = −0.15, *p* < 0.01), and academic disaffection (r = 0.14, *p* < 0.01), which partially supports Hypothesis 1. In addition, the incidence of parental burnout in the studied sample was 5.36%, which also coincides with the findings of Wang et al. [40].

### 3.3. Hypotheses Testing Using Structural Equation Modeling

Structural equation modeling was employed to examine the impact of parental burnout in primary caregivers on adolescents’ overall psychological development and the mediating role of negative parenting styles. The model showed acceptable goodness of fit: χ^2^ = 86.49; df = 29; CFI = 0.93; RMSEA = 0.09; and SRMR = 0.02. The results of the path analysis are shown in Figure 2.

Figure 2 shows that parental burnout in primary caregivers is positively related to their negative coping styles (β = 0.18, *p* < 0.01); therefore, Hypothesis 2 was supported. The negative coping styles expressed by primary caregivers are negatively related to the positive psychological development of adolescents (β = −0.53, *p* < 0.001) and positively related to adolescents’ negative psychological development (β = 0.75, *p* < 0.001). Bootstrapping (n = 5000) was used to further examine the mediating effects. The results showed that negative coping styles mediated the relationship between parental burnout and positive psychological development (β = −0.03, *p* = 0.03, 95% CI [−0.07, −0.01]), and between parental burnout and negative psychological development (β = 0.04, *p* = 0.03, 95% CI [0.01, 0.10]); therefore, Hypothesis 3 was supported.

We controlled for parents’ and adolescents’ sex in the model, and the results showed no meaningful change. As controlling for too many variables would decrease the power of the analysis [47], we only report the original results of the analysis.

## 4. Discussion

This study examined primary caregivers’ parental burnout on adolescents’ development and the mediating role of negative parenting styles. Specifically, the positive and negative aspects of psychological adaptation, friendship quality, and academic engagement were selected as indicators of adolescent development; psychological control and parental neglect were selected as negative parenting styles. From the time-lagged design and multi-source data, the results generally supported the hypothesized model. According to Wang et al. [40], the incidence of fathers’ parental burnout was 1.5–12.83%. Further, 5.36% of parents experienced parental burnout. Considering the large population base in China, the risk of parental burnout is still notable.

### 4.1. Parental Burnout and Adolescents’ Development

As the impact of parenting behavior during adolescence persists and influences behaviors into adulthood, it is critical to explore the effects of parental burnout on adolescents’ development [48]. Prior studies indicate that burned-out parents are inclined to exhibit more neglectful and violent behaviors toward their children [7,14], potentially resulting in adverse consequences to children’s development. Particularly, parental burnout can affect adolescents’ mental health [29], lead to problematic behaviors [9], academic burnout [11], social distress and academic performance [12]. In line with prior studies, this study demonstrated that parental burnout could negatively predict positive adolescent development and positively predict negative adolescent development, especially in academic engagement and academic disaffection. The results not only corresponded to previous research but also provided new evidence for the negative effects of parental burnout on adolescents’ development.

Notably, the direct effects of parental burnout on adolescents’ psychological adaptation and friendship quality were not significant. This may be because Chinese parents emphasize their children’s academic performance [11], paying less attention to their psychological adaptation and friendship quality. Therefore, the direct influence of parental burnout is relatively weak. Another possibility is that the influence of parental burnout on adolescents’ psychological adaptation and friendship quality may be indirect. Future research needs to further explore the relationship between parental burnout and adolescents’ development.

### 4.2. Parental Burnout and Negative Parenting Style

Parents who adopt positive parenting styles are less prone to symptoms of parental burnout [2]; however, burned-out parents may have a series of parenting style changes; for instance, high-level parental burnout is likely to reduce emotional investment and increase neglect and violent behavior in them [7,14]. Moreover, it is significantly positively correlated with psychological control [34]. In line with prior studies, the current results also showed that parental burnout was positively associated with negative parenting style.

Specifically, parental burnout can prolong parents’ negative mental state, thereby making it difficult for them to obtain a sense of accomplishment from parenting activities. They may try to escape their parenting responsibilities, and hence, consciously ignore adolescents’ physical and emotional needs [7,14]. Conversely, parents have to continue to fulfill their parenting responsibilities because they cannot completely separate themselves from their parental roles, while the legal and moral aspects require them to be responsible and obligated to raise their children. Faced with the continuous consumption of parenting resources, parents tend to interact with their children through psychological control [34].

### 4.3. Mediating Role of Parenting Styles

Parenting style produces direct predictive effects on the physical and mental development of adolescents [18]. A negative parenting style may increase the internalization and externalization of adolescent problem behaviors [15]. In line with prior studies, this study examined negative parenting styles as mediators of the relationship between parental burnout and positive/negative adolescent development, and the indirect effects were significant.

Owing to chronic emotional exhaustion, parents may change their parenting styles to avoid losing more resources [2]. For adolescents, parental neglect and psychological control can aggravate parent–child conflicts [34], which may further hinder the formation of the adolescent’s self-identity and increase the likelihood of psychological problems in them [49].

Additionally, our results remained substantially unchanged when controlling for the sex of both the parents and adolescents. Given our emphasis on parenting roles, we posit that the primary caregiver, typically the one with a more meaningful impact on the children, plays a more crucial role than the secondary caregiver. This perspective may partially explain the result showing that the influence of parents on their children was not significantly affected by the parents’ sex; however, it is noteworthy that the traditional familial concept of “The man goes out to work while the woman looks after the house” still persists in Chinese parenting practice. This suggests that further research is warranted to explore the distinct effects of fathers’ and mothers’ parental burnout on children in diverse cultures. This exploration aims to demonstrate the differential impact of parental burnout on children when considering fathers versus mothers, and whether parental burnout in the primary caregiver, irrespective of sex, has uniform effects on children across cultures.

### 4.4. Theoretical and Practical Implications

While parental activities require the participation of both fathers and mothers, they may assume different roles in different families. Primary caregivers play a more vital role in parenting than secondary caregivers, regardless of whether they are fathers or mothers; therefore, this study identified the primary caregivers and examined the effects of parental burnout on adolescent development. Clarifying this concept could promote more accurate research on the consequences of parental burnout. Further, this study examined the impact of parental burnout on adolescent development from multiple perspectives. The findings underscore the significance of parents not only preventing themselves from experiencing burnout but also refraining from adopting negative parenting styles if they do find themselves in a state of burnout. This proactive approach is essential to minimize any adverse effects on their children.

### 4.5. Limitations and Future Directions

This study has several limitations. First, although the time-lag design was used for data collection thrice with an interval of one month, the results can only indicate the impact of parental burnout on adolescent development in the short term. Future studies should consider longitudinal designs with different time intervals and compare the differences in the impact of parental burnout on adolescent development in different phases. In addition, the data were collected from different sources, which could decrease common method bias; however, a multi-wave design to collect all variables in each measurement time could provide better control for the model. Further studies should employ a more rigorous design. Second, although this study proposed the concept of “primary caregivers” and added the option of “others” to describe parents in the data survey, all responses that included “others” were excluded because the answers could not be matched between adolescents and parents. As society continuously evolves, grandparents, uncles, and other relatives in some families may also function as primary caregivers. Hence, future research should improve and refine the group of “primary caregivers” in families. Third, we focused on the indirect effects of parental burnout on adolescents’ development through negative parenting styles, which ignored the possibility of parental burnout on adolescents’ development through a decrease in positive parenting styles. Further studies should examine the effects of parental burnout on adolescents’ development from a more comprehensive perspective that examines both positive and negative parenting styles.

## 5. Conclusions

This study focused on the effects of primary caregivers’ parental burnout on adolescents’ development and the mediating role of negative parenting styles. The results demonstrated that burned-out parents are likely to change their parenting styles, negatively affecting adolescents’ development in psychological adaptation, friendship quality, and academic engagement. Therefore, parents who are already grappling with parental burnout should refrain from adopting negative parenting styles or behaviors to mitigate the adverse effects on their children.

## Figures and Tables

**Figure 1 behavsci-14-00161-f001:**
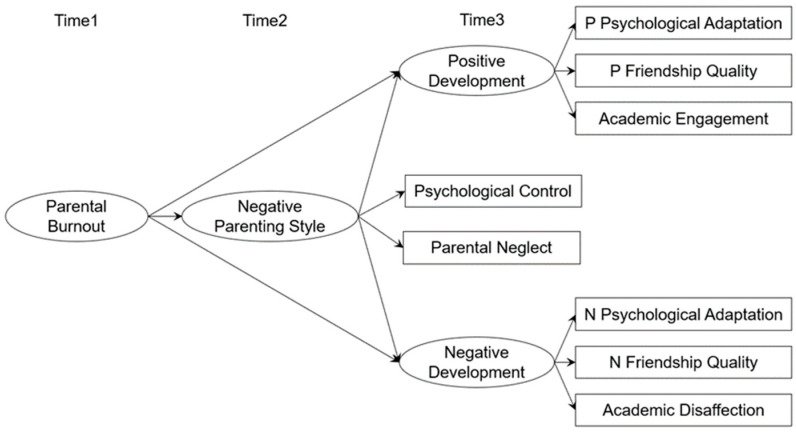
Theoretical framework of the mediating role of negative parenting style in the relationship between parental burnout and adolescent development. Note: P Psychological Adaptation = positive psychological adaptation; N Psychological Adaptation = negative psychological adaptation; P Friendship Quality = positive friendship quality; N Friendship Quality = negative friendship quality.

**Figure 2 behavsci-14-00161-f002:**
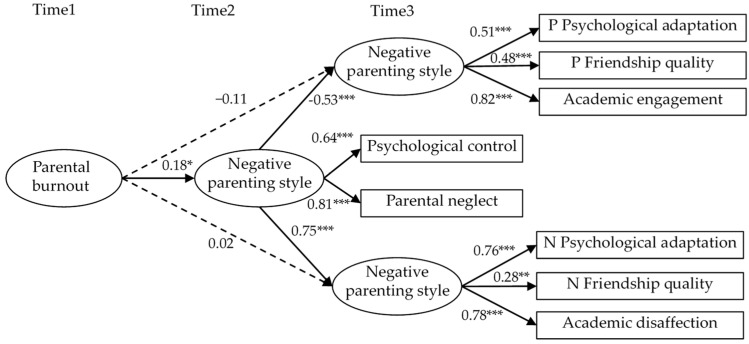
Result of analyzed model. Note: solid dark arrows indicate hypothesized paths that are significant. Dashed dark arrows indicate hypothesized paths that were estimated but are not significant. The path coefficients are standardized. * *p* < 0.05, ** *p* < 0.01, *** *p* < 0.001.

**Table 1 behavsci-14-00161-t001:** Descriptive statistics and correlations between study variables.

	M (SD)	1	2	3	4	5	6	7	8
1.T1 Parental burnout	1.45 (0.80)								
2.T2 Psychological control	2.55 (0.89)	0.19 **							
3.T2 Parental neglect	0.79 (0.70)	0.12 ^†^	0.52 **						
4.T3 P Psychological adaptation	1.59 (0.38)	−0.10	0.02	−0.26 **					
5.T3 N Psychological adaptation	0.51 (0.27)	0.09	0.44 **	0.48 **	−0.30 **				
6.T3 P Friendship quality	2.78 (0.76)	−0.07	−0.16 **	−0.25 **	0.41 **	−0.33 **			
7.T3 N Friendship quality	0.82 (0.69)	0.04	0.24 **	0.15 **	0.05	0.28 **	0.05		
8.T3 Academic engagement	3.19 (0.55)	−0.15 *	−0.23 **	−0.41 **	0.41 **	−0.46 **	0.35 **	−0.13 *	
9.T3 Academic disaffection	1.58 (0.56)	0.14 *	0.31 **	0.45 **	−0.27 **	0.58 **	−0.24 **	0.24 **	−0.60 **

Note: P Psychological Adaptation = positive psychological adaptation; N Psychological Adaptation = negative psychological adaptation; P Friendship Quality = positive friendship quality; N Friendship Quality = negative friendship quality. T1, Time 1; T2, Time 2; T3, Time 3. ^†^
*p* < 0.10, * *p* < 0.05, ** *p* < 0.01.

## Data Availability

The datasets generated for this study are available upon request to the corresponding author.

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
