# Peer review of "Parental Burnout, Negative Parenting Style, and Adolescents’ Development"

_behavsci, 2024, doi:10.3390/bs14030161_

Round 1

Reviewer 1 Report

Comments and Suggestions for Authors

1.Please add the recent references in the introduction.

2.Please explain the Parental Burnout deeply.

3. Please explain the relationship between  parental burnout and adolescents’ development in the discussion.

4. Please polish the manuscipt.

Comments on the Quality of English Language

Please polish the manuscipt.

Author Response

  1. Please add the recent references in the introduction.

Thank you for your review on our manuscript entitled “Parental burnout, negative parenting style and adolescents’ development” (behavsci-2826456). We strongly agree with your comments on the limitations. We have revised the manuscript according to your suggestions and the revised section was highlight in yellow color.

Thank you for your comments. Combined with your and other reviewer’ suggestion, we have rewritten the introduction section and added more recent references.

  1. Please explain the Parental Burnout deeply.

Thank you for your comments. We have added the explanation of each symptom of parental burnout.

  1. Please explain the relationship between parental burnout and adolescents’ development in the discussion.

Thank you for your comments. We have added the explanation of the relationship between parental burnout and adolescents’ development in the discussion section (line 376-381).

  1. Please polish the manuscript.

Thank you for your suggestion. We have read the manuscript carefully, and asked the editage company to help us do the proofreading work.

Reviewer 2 Report

Comments and Suggestions for Authors

This is an interesting article that might be improved by further revision.

In the article, at the outset please indicate the context and aims of the research. What practical purpose does this study have and what does it aim to achieve? This must also be addressed in concluding material.

Especially in the opening section, the material can be streamlined somewhat, offering an indication of the previous gaps in knowledge that are relevant to this study in particular. There is a degree of confusion regarding the actual focus of the piece until you present your two hypotheses.

There is no indication of the ethical basis for this study. How did you gain ethical approval and what ethical concerns are raised in a study such as this? 

It is unclear what the age range of the study is. The terms children and adolescents seem to be used interchangeably, and as the data is gained from participants in a primary school, it would be helpful to know what their ages are. Again, it is important to provide some detail on the ethical issues concerning research with children and how you have mitigated these concerns. 

Finally, in the conclusion, please indicate how the study is relevant to your own context, but beyond to parenting in other countries.

Comments on the Quality of English Language

The quality of English is satisfactory. However, there are a high number of spelling and typing errors as well as sentence structure issues. Please correct these in your revision.

Author Response

This is an interesting article that might be improved by further revision.

Thank you for your review on our manuscript entitled “Parental burnout, negative parenting style and adolescents’ development” (behavsci-2826456). We strongly agree with your comments on the limitations. We have revised the manuscript according to your suggestions and the revised section was highlight in yellow color.

In the article, at the outset please indicate the context and aims of the research.

Thank you for your comments. Combined with your and other reviewer’ suggestion, we have rewritten the introduction and added the context and aims of the research at the beginning of the manuscript.

What practical purpose does this study have and what does it aim to achieve? This must also be addressed in concluding material.

Thank you for your comments. We have added the practical purpose to the conclusion section.

Especially in the opening section, the material can be streamlined somewhat, offering an indication of the previous gaps in knowledge that are relevant to this study in particular. There is a degree of confusion regarding the actual focus of the piece until you present your two hypotheses.

Thank you for your comments. We have rewritten the introductory section and removed redundancies.

There is no indication of the ethical basis for this study. How did you gain ethical approval and what ethical concerns are raised in a study such as this? 

Thank you for your comments. We have added the ethical information of the study.

It is unclear what the age range of the study is. The terms children and adolescents seem to be used interchangeably, and as the data is gained from participants in a primary school, it would be helpful to know what their ages are. Again, it is important to provide some detail on the ethical issues concerning research with children and how you have mitigated these concerns. 

Thank you for your comments. We have added the age information to the manuscript.

We are sorry about our writing. We are trying to use CHILD to represent their offspring and ADOLESCENTS to represent our sample. We have rewritten the relevant section to minimize misunderstandings.

Finally, in the conclusion, please indicate how the study is relevant to your own context, but beyond to parenting in other countries.

Thank you for your comments. We have added the discussion of culture context to the conclusion section.

Reviewer 3 Report

Comments and Suggestions for Authors

INTRODUCTION

Much work is needed in the introduction.  The introduction to the paper is currently very disjointed and unclear.  The goal in the intro is to not only educate the reader on the concepts being studied, but to also make a clear argument for how your research fits into the larger academic narrative on the topic.  As a reader, I found myself confused by the organization of the information, and many aspects of your work are still unclear to me.

I believe you are making the general argument that parenting behaviors/styles impact adolescent developmental outcomes and that when parents experience burnout, adolescent developmental outcomes can be impacted.  If this is, in fact, the general argument, then organize the introduction in such a way that brings the reader from the general to the more specific.  A coherent discussion of the literature might encompass the following subsections:

(1)  A discussion of how parenting and development are related.  Included in this discussion could be how positive parenting styles might lead to positive developmental outcomes and negative parenting styles might lead to negative developmental outcomes.

(2)  A discussion about variables that might impact parenting styles—in this case, parental burnout.  Included in this section would be a discussion of what parental burnout is and how it impacts parenting behavior.

(3)  A discussion of how all of these variables relate together. What is currently known in the research about how parental burnout, parenting behaviors, and adolescent development relate?

Then, you can move toward how your study will add to this information.  Right now, you have interwoven aspects of how your study defines things like social adaptation and what frameworks and models you plan to use with background information. This creates a lack of clarity. Intertwining the information about your study with the background content needed in an introduction confuses the issue and makes it difficult for the reader to understand how your study fits into the larger discussion about these concepts within the field as a whole.

Other general questions comments from the introduction:

Lines 58 and 59, the two sentences should be merged.

Line 60: the title of the work is not needed—that can be obtained from the reference section.  Just state that the indicators of academic achievement and social adaptation were chosen from the work of Dong & Lin (2011).  You also provide the title later in the introduction and that is not needed.

More information is needed about the indicators found by Dong and Lin (2011).  The summary provided in lines 60-66 is a bit unclear.  Can you summarize the framework?

I like the focus on the primary caregiver, as there can be differential effects from those that take on the primary role for daily parenting tasks.  However, see my further comments about this for the results section.

Why the choice of psychological control and parental neglect as indicators of negative parenting?  It’s not made clear why these aspects were chosen.

In Line 126, you refer to “this study.”  Is that a reference to the Dong and Lin (2011) study or your study?  If it’s your study, how does this relate to the work of Dong and Lin?  That has not been made clear.  This confusion speaks to the more general comments above.  First discuss background, then show how your study fits or adds to what is known.

MATERIALS AND METHODS

Why the separation into three points in time for data collection if different information was collected at each time point?  This is unclear and should be discussed.

RESULTS & DISCUSSION

Perhaps organize your results section to include a specific sub-section for each of your hypotheses.

Given that you use both mothers and fathers in your sample (based on who is identified as the primary caregiver), I think it’s important that you controlled for parents’ sex (and the sex of the adolescent, as well).  I would like to see more discussion of this in the intro/discussion.  Given that men and women tend to be socialized differently for the parenting role—with more of the caregiving “pressure” put on women, it might be worthwhile to include acknowledgement of this throughout. 

Parts of your discussion and introduction take a negative view of the adolescent life stage.  For instance, lines 363-364 discuss children/adolescents as being “relatively immature.” I would switch the narrative and instead refer to parenting at this stage as having its own unique challenges, just as parenting does at other stages of the lifespan.

Comments on the Quality of English Language

Minor editing of English language required

Author Response

INTRODUCTION

 Much work is needed in the introduction.  The introduction to the paper is currently very disjointed and unclear.  The goal in the intro is to not only educate the reader on the concepts being studied, but to also make a clear argument for how your research fits into the larger academic narrative on the topic.  As a reader, I found myself confused by the organization of the information, and many aspects of your work are still unclear to me.

Thank you for your review on our manuscript entitled “Parental burnout, negative parenting style and adolescents’ development” (behavsci-2826456). We strongly agree with your comments on the limitations. We have revised the manuscript according to your suggestions and the revised section was highlight in yellow color.

I believe you are making the general argument that parenting behaviors/styles impact adolescent developmental outcomes and that when parents experience burnout, adolescent developmental outcomes can be impacted.  If this is, in fact, the general argument, then organize the introduction in such a way that brings the reader from the general to the more specific.  A coherent discussion of the literature might encompass the following subsections:

 (1)  A discussion of how parenting and development are related.  Included in this discussion could be how positive parenting styles might lead to positive developmental outcomes and negative parenting styles might lead to negative developmental outcomes.

(2)  A discussion about variables that might impact parenting styles—in this case, parental burnout.  Included in this section would be a discussion of what parental burnout is and how it impacts parenting behavior.

(3)  A discussion of how all of these variables relate together. What is currently known in the research about how parental burnout, parenting behaviors, and adolescent development relate?

Then, you can move toward how your study will add to this information.  Right now, you have interwoven aspects of how your study defines things like social adaptation and what frameworks and models you plan to use with background information. This creates a lack of clarity. Intertwining the information about your study with the background content needed in an introduction confuses the issue and makes it difficult for the reader to understand how your study fits into the larger discussion about these concepts within the field as a whole.

Thank you for your comments. We have rewritten the introduction section according to your suggestion.

Other general questions comments from the introduction:

 Lines 58 and 59, the two sentences should be merged.

 Thank you for your comments. We have combined these two sentences.

Line 60: the title of the work is not needed—that can be obtained from the reference section.  Just state that the indicators of academic achievement and social adaptation were chosen from the work of Dong & Lin (2011).  You also provide the title later in the introduction and that is not needed.

 Thank you for your comments. We have removed the relevant section.

More information is needed about the indicators found by Dong and Lin (2011).  The summary provided in lines 60-66 is a bit unclear.  Can you summarize the framework?

 Thank you for your comments. We have removed the detail from line 60-66. And  added more detail of the framework to the later in the introduction section (line 122-128.

I like the focus on the primary caregiver, as there can be differential effects from those that take on the primary role for daily parenting tasks.  However, see my further comments about this for the results section.

 Why the choice of psychological control and parental neglect as indicators of negative parenting?  It’s not made clear why these aspects were chosen.

Thank you for your comments. We have added the reason why we chose the two variables as mediators(Line 72-76) .

In Line 126, you refer to “this study.”  Is that a reference to the Dong and Lin (2011) study or your study?  If it’s your study, how does this relate to the work of Dong and Lin?  That has not been made clear.  This confusion speaks to the more general comments above.  First discuss background, then show how your study fits or adds to what is known.

 Sorry about our writing. We have revised the sentence to “the present study”(Line 130).

MATERIALS AND METHODS

 Why the separation into three points in time for data collection if different information was collected at each time point?  This is unclear and should be discussed.

Thank you for your comments. The data was collected at three time points mainly for decrease the common method bias, meanwhile, this allows us better causal inferences than collecting data at one point in time.

RESULTS & DISCUSSION

 Perhaps organize your results section to include a specific sub-section for each of your hypotheses.

Thank you for your comments. We have added “Hypothesis were supported” to the corresponding results. As we adopted structural equation modeling to tested all hypotheses at same time, the main result for hypothesis examination was written in one paragraph.

reorganized the result section.

Given that you use both mothers and fathers in your sample (based on who is identified as the primary caregiver), I think it’s important that you controlled for parents’ sex (and the sex of the adolescent, as well).  I would like to see more discussion of this in the intro/discussion.  Given that men and women tend to be socialized differently for the parenting role—with more of the caregiving “pressure” put on women, it might be worthwhile to include acknowledgement of this throughout. 

Thank you for your comments. Given our emphasis on parenting roles in this study, we posit that the primary care-giver, typically the one with a more significant impact on the children, plays a crucial role than the secondary caregiver. This perspective may partially explain the result that the influence of parents on children was not found to be significantly affected by parents’ gender. We have added the discussion of gender difference to the discussion section (line 426-438).

Parts of your discussion and introduction take a negative view of the adolescent life stage.  For instance, lines 363-364 discuss children/adolescents as being “relatively immature.” I would switch the narrative and instead refer to parenting at this stage as having its own unique challenges, just as parenting does at other stages of the lifespan.

Thank you for your comments and sorry about our writing. We have removed the relevant sentence.

Round 2

Reviewer 2 Report

Comments and Suggestions for Authors

Thankyou for attending to my requests in the first review.

To finalise the revision process please attend to the following:

* In the first section of the paper, there are a number of issues with punctuation and sentence structure. It might be worth employing your colleague again to help strengthen this as some work is required to ensure the writing is at a suitable level for academic publication.

* In the short paragraph on ethical approval, please also indicate how you gained consent from the participants. 

Comments on the Quality of English Language

There is some improvement here but as stated in above, please pay attention to the Introduction (section 1)

Author Response

Thank you for attending to my requests in the first review.

To finalize the revision process please attend to the following:

* In the first section of the paper, there are a number of issues with punctuation and sentence structure. It might be worth employing your colleague again to help strengthen this as some work is required to ensure the writing is at a suitable level for academic publication.

* In the short paragraph on ethical approval, please also indicate how you gained consent from the participants. 

Thank you for reviewing our manuscript. We are very sorry about our writing. We have made further modifications and invited Editage company to conduct proofreading. In addition, regarding the consent, written informed consent has been obtained from the participants, and participants need to check the box of “Please check this box if you agree that the data can be used for academic research and publication.” We have added the above content to the main text (Line 253-257).